# Subfamily Anischiinae (Coleoptera: Eucnemidae) in Early Cretaceous of Northeast China

**DOI:** 10.3390/insects12020105

**Published:** 2021-01-26

**Authors:** Haolun Li, Huali Chang, Jyrki Muona, Yanchen Zhao, Dong Ren

**Affiliations:** 1College of Life Sciences and Academy for Multidisciplinary Studies, Capital Normal University, Xisanhuanbeilu 105, Haidian District, Beijing 100048, China; 2190802111@cnu.edu.cn (H.L.); zyc910312@outlook.com (Y.Z.); 2Henan Geological Museum, Jinshuidonglu 18, Zhengdongxin Dsitrict, Zhengzhou 450016, China; changxinyin07@gmail.com; 3Entomology Team, Zoology Unit, Finnish Museum of Natural History, FIN-00014 University of Helsinki, 00100 Helsinki, Finland

**Keywords:** Coleoptera, Eucnemidae, Anischiinae, new taxa, fossils, Jehol Biota, Yixian Formation, Cretaceous, China, angiosperms, gymnosperms, evolutionary specialization

## Abstract

**Simple Summary:**

A new false click-beetle (Eucnemidae) belonging to subfamily Anischiinae, formerly not known as fossils, extends the age of this small extant group 125 million years back. The structural specializations of the false click-beetle larvae are discussed. On the basis of this new piece of paleontological evidence, it is shown that these insects have changed their host preference from gymnosperms to angiosperms at least twice. It also indicates that the beetle larvae can switch their favored development substrate by adapting in different ways, either with or without morphological specialization.

**Abstract:**

*Rheanischia* new genus, type species *Rheanischia brevicornis* new species (Eucnemidae, Anischiinae) is described from the Lower Cretaceous of Liaoning, China. The presence of this species in early Cretaceous deposits provides new insight into the evolution of basal lignicolous Eucnemidae clades. Both Anischiinae and Palaeoxeninae species diversified in a world dominated by gymnosperms, before the main radiation of angiosperms. More than 95% of modern eucnemid larvae have a *Palaeoxenus*-type highly modified head structure, but contrary to the *Palaeoxenus* larva, they develop in angiosperm wood. Anischiinae utilize angiosperms as well, but their head capsule shows no such modifications. These facts prove that highly specialized morphological features do not offer definite proof of similar way of life in the distant past, nor should non-modified structures be taken as proof for another kind of substrate choice. Eucnemidae have invaded angiosperms with two quite different morphological adaptations. This fact may have implications for the evolution of all clicking elateroids.

## 1. Introduction

The superfamily Elateroidea includes four families with species possessing a defensive “clicking” mechanism based on an advanced promesothoracic locking structure [1]. The largest of these, the click-beetles (Elateridae), is one of the major beetle radiations with more than 10,000 extant species [2] and a rich fossil history [3].

The second largest family, false click-beetles (Eucnemidae), includes more than 1700 extant species [4]. Their fossil history remained quite obscure until Muona [5] reviewed the early records and reported 18 genera with 34 species from the Baltic amber. Most of these were described from Eocene samples, but five genera (*Microrhagus* Dejean, *Sieglindea* Muona, *Erdaia* Muona, *Euyptychus* LeConte, *Fornax* Laporte) were reported both from the Miocene and Eocene.

The first Mesozoic eucnemids were described from lower Cretaceous China [6] and upper Jurassic Australia [7]. Otto [8] and Li et al. [9] added one species each from the late Cretaceous Burmese amber. Our knowledge of the Mesozoic fauna was greatly expanded when Muona et al. [10] showed that the previously described Chinese elaterid fossils included one Jurassic and six early Cretaceous eucnemids, all of them belonging to extant monotypic subfamilies (Pseudomeninae Muona, Schizophilinae Muona, and Palaeoxeninae Muona).

Finally, Muona [11] reported 15 genera and 22 species of eucnemids from the late Cretaceous Burmese amber deposits. Of these, eight genera and 20 species were described as new. One genus had Eocene and Miocene Baltic amber species as well as extant ones (*Euryptychus* LeConte), one genus (*Sieglindea* Muona) was previously known from Eocene and Miocene Baltic amber, and three genera had extant species (*Epiphanis* Eschscholtz, *Jenibuntor* Muona, *Myall* Muona). In addition to the subfamily Palaeoxeninae, these samples included the first appearance in fossil record of the subfamilies Eucnemiinae Eschscholtz, Melasinae Fleming, and Macraulacinae Fleutiaux—the three dominant Eucnemidae clades today.

Here, we report the first fossil species belonging to the eucnemid subfamily Anischiinae Fleutiaux. The placement of the fossil follows the guidelines presented in Muona et al. [10].

## 2. Materials and Methods

This study is based on two specimens from Liaoning, housed in the fossil insect collection of the Key Laboratory of Insect Evolution and Environmental Changes, College of Life Science, Capital Normal University, Beijing, China. The specimens were examined using a Nikon SMZ25 dissecting microscope and the line drawings were produced with the software Adobe Illustrator 17.0. Photographs were acquired with a Nikon SMZ25 Digital Camera. Body length was measured along the midline from the anterior edge of the mandible to the apex of the abdomen, and the width was measured across the broadest part of the elytra. The length of the pronotum was measured along the midline; the width was measured across the broadest part at its posterior angles.

The specimens were collected near Chaomidian Village, Liaoning Province, China. This site is part of the famous Jehol Biota [12], belonging to the Yixian Formation stratigraphically, consisting mainly of lacustrine sediments intercalated with volcanoclastics [13,14,15]. Paleobotanical data from fossil spores, pollen, and plants indicate a rather warm and humid climate at that time [12,16,17,18,19]. The age of the strata is regarded as Early Cretaceous (Hauterivian to Aptian in age) [15,20,21].

## 3. Results

Systematic palaeontologyClass: Insecta Linnaeus, 1758;Order: Coleoptera Linnaeus, 1758;Superfamily: Elateroidea Leach, 1815;Family: Eucnemidae Eschscholtz, 1829;Subfamily: Anischiinae Fleutiaux, 1936;

Elateroidea *sensu stricto* synapomorphy: promesothoracic clicking mechanism present.

Elateroidea *sensu stricto* diagnostic other features: form elongated, flattened; metacoxal plates enlarged metaventrite without transverse suture (Figure 1 and Figure 2).

Eucnemidae synapomorphies: elytra with sutural and lateral striae apically with enlarged excretory pores (Figure 3A); pedicel attached to scape subapically (Figure 3B).

Eucnemidae other diagnostic feature: antennae with three last flagellomeres enlarged, forming a loose club (Figure 3B).

Anischiinae synapomorphies: pronotal disc with pair of lateral grooves with carinate edge, extending anteriorly from interlocking cavities at base to middle of disc (Figure 1, Figure 2 and Figure 3); pronotal hind edge with well-developed interlocking device, including pair of deep lateral cavities for receiving paired processes on anterior edges of elytra, anterior edge of each elytron carinate, and produced at middle to form lobe fitting into cavity on pronotum (Figure 1).

Anischiinae other diagnostic characters: flagellomeres flattened, stout, bulbous in profile, antennae apically abruptly expanded to form weak, three-segmented club (Figure 3B); labrum attached underneath frontoclypeal region, faintly visible (Figure 1A and Figure 3B).

Genus: *Rheanischia* Li, Chang and Muona new genus.

Type species: *Rheanischia brevicornis* Li, Chang and Muona new species.

Etymology: a combination of *Rhea*, one of the female titans in Greek mythology, and *Anischia*, the type-genus of the subfamily, indicating that species in this genus are much larger than previously known Anischiinae.

### 3.1. Diagnosis

The promesothoracic clicking mechanism, elytra with apical excretory pores, and the subapically attached pedicel place this genus in Eucnemidae. Within Eucnemidae, it shares the unique pronotal cavities and the interlocking system between pronotum and elytra with the extant genus *Anischia* (Figure 1 and Figure 4A). The stout clavate antennae and prominent postcoxal lines on abdominal sternite one are further features typical of that genus (Figure 4B). *Anischia* differs from *Rheanischia* by being much smaller, having a more rounded body, lacking elytral striae except the sutural one, having prominent vestiture (Figure 4A) acutely angled coxal lines on first visible ventrite, and having hardly visible metacoxal plates (Figure 4B).

### 3.2. Description

Body large, flat, and wide (Figure 1 and Figure 2). Antennae stout, relatively short, scape short, flagellomeres robust to moderately slender, seven to nine clavate (Figure 3B). Head prognathous, wide, subtriangular, mandibles elongated, exposed, fairly large, labrum feebly exposed under the somewhat protruding frontoclypeal region, eyes medium-sized, prominent (Figure 3B). Pronotum with acute hind angles, slightly diverging craniad, with complete lateral carinae separating disk and hypomera, disk with laterally defined cavities extending over half of the basal length, caudal margin forming interlocking system with base of elytra (Figure 1 and Figure 2). Elytra with well-developed sharp sutural striae, other striae distinct but fine, with minute punctures, sutural striae and lateralmost striae apically with excretory punctures (Figure 1). Hypomera without antennal grooves, meso- and meatathoracic sclerites plesiomorphic, metaventrite wide and extensive, without transverse suture, procoxal cavities nearly circular, moderately spaced, mesocoxal cavities moderately spaced with complete coxal line, metacoxal femoral plates large, subtriangular, strongly narrowing laterad, abdominal sternite one with prominent coxal lines (Figure 2). Male genitalia exposed in holotype, but vaguely visible, dorsoventreally flattened, appearing to be simply trilobed and having a complete subgenital plate.

Legs long and slender, femora simple, slender, tibiae long, metatibiae with rows of spines, tarsi slender (Figure 1).

*Rheanischia brevicornis* Li, Chang and Muona new species.

Etymology: *brevicornis* refers to stout antennae.

Holotype: male, in Key Laboratory of Insect Evolution and Environmental Changes, College of Life Science, Capital Normal University, Beijing, China, CNU-COL-LB2018344. Dimensions: Body length, 15.28 mm; head width, 1.94 mm; pronotum width, 4.29 mm; length 2.97 mm; elytron length, 9.79 mm; combined elytral width, 5.20 mm.

Paratype: sex unknown, CNU-COL-LB2018313. The specimen has a major part of the abdomen and elytra broken off. Deposited in the same collection as the holotype.

### 3.3. Diagnosis

Within Anischiinae characterized by the wide, parallel-sided elaterid-like body, short, stout antennae, and widely curved coxal lines on first visible ventrite.

### 3.4. Description

Antennae stout, scape oval, longer and thicker than flagellomere 1, pedicel slightly longer than wide, length of flagellomere 2, 1.4X its width, 3–6 about equal, longer than wide, subconical, flagellomeres 7–8 more dilated, distinctly wider than long, flagellomere 9 slenderer than 7 or 8, oval, small (Figure 3B). Pronotal width 1.4X its length, widest at hind angles, very slightly arcuate or straight anteriorly, close to parallel in basal fifth, but narrowing evenly anteriorly. Elytra with distinct, fine striae, interstices convex at apex, all striae visibly punctate, large apical excretory punctures concentrated on sutural striae and lateral ones.

## 4. Discussion

The most striking external feature in extant *Anischia* species is the structure of the pronotum. The interlocking system between pronotum and elytra as well as the deep, laterally defined grooves on pronotal disk are unique within Eucnemidae. Similar structures are found in Elateridae: Cardiophorinae and Negastriinae [22], but these taxa lack synapomorphies defining Eucnemidae: pedicel attached subapically to scape, labrum attached underneath the fronotclypeal region, and larval legs extremely reduced. Cardiophorinae and Negastrinae share an apomorphy not found in *Anischia*, female bursa with a structure called colleterial glands—a putative synapomorphy for some or all Elateridae [22], but see also [23]. The actual nature of these gland-like features is unclear, however.

Douglas (pers. comm.) has pointed out to us that some Cardiophorinae elaterids have enlarged punctures apically on elytra, a feature characterizing the majority of Eucnemidae. Douglas [23] coded this character (his number 90) as missing for an unidentified Central American *Anischia* species. We have checked several *Anischia* species and have been able to verify that *Anischia mexicana* Fleutiaux (Mexico), *Anischia boliviana* Fleutiaux (Bolivia), *Anischia ruandana* (Basilewsky) (Ivory Coast), and several undescribed species from Panama, Colombia, Brazil, and Peru have large excretory punctures apically, located in longitudinal sutural depressions, but *Anischia bicolor* Lawrence (New Caledonia), *Anischia kuscheli* Lawrence (New Caledonia), *Anischia stupenda* Fleutiaux (New Guinea), and four undescribed species from Vanuatu, Fiji, and the Philippines have at most the shallow sutural apical depressions on elytra. This may show a further evolutionary development of this character. Whether this reflects the presence of two separate clades remains to be seen.

The overall appearance of the large extinct *Rheanischia* species (Figure 1) is strikingly different from that of the extant *Anischia* species (Figure 4A). The systematic position of *Anischia* in Eucnemidae was firmly established with a total evidence approach only quite recently [24]. This result was confirmed in an analysis with molecular sampling as well [25]. The discovery of *Rheanischia* shows that the seemingly odd eucnemid, *Anischia*, is actually a derived member of an ancient clade present in early Cretaceous, today surviving as a small pantropical, morphologically modified group.

Excluding the Anischiinae synapomorphies from consideration for a moment, *Rheanischia* becomes a fairly “normal” basal eucnemid. The abdomen appears to be connate; body form is flattened and elaterid-like; antennae have three enlarged apical flagellomeres; elytra have striae; metacoxal plates are large and strongly narrowing laterad; metatibiae have spine-combs and the male aedeagus appears to be trilobed, dorsoventrally flattened. All these features suggest a position close to *Palaeoxenus* Horn, another extant eucnemid clade known from the Yixian fauna as well [6]. This relationship is basically what Lawrence et al. suggested [24]. Muona and Teräväinen [26], however, were not able to find a placement for *Anischia* within Eucnemidae on the basis of its unusual larva. The discovery of Anischiinae fossils with a set of plesiomorphic elateroid features combined with the Anischiinae synapomorphies offered support for the earlier hypothesis [24], and clarified the exact position of Anischiinae within Eucnemidae. The polymorphism of apical elytral excretory punctures in extant species is exceptional, but as stated in Muona and Teräväinen [10], this feature has been lost secondarily in some derived Macraulacini as well.

The *Anischia* larval structures, including large endodont mandibles, head without lateral teeth, and segmented legs, should be accepted as plesiomorphic features, and the whole “derived eucnemid clade” (“DE” in [26]) should be regarded as a sister-group to it. The informal group DE consists of Palaeoxeninae and its sister group, all other derived lignicolous eucnemids, i.e., Eucneminae, Macraulacinae, Melasinae and Phlegoninae. The larval structures are the key synapomorphies uniting those beetles: exodont mandibles, head capsule with highly developed lateral structures, and total loss of segmented legs [26]. The evolutionary sequence for the lignicolous Eucnemidae *sensu* [26], including Anischiinae is thus: (Pseudomeninae) ((Schizophilinae) ((Anischiinae ((Palaeoxeninae) ((Phlegoninae) (Melasinae, Eucneminae, Macraulacinae)))))).

If the present host choices of these clades are seen as reflecting the original ones, this hypothesis means that the eucnemids invaded either angiosperms or gymnosperms multiple times. The larva of *Pseudomenes bakewelli* (Bonvouloir) has been collected from “rotten log” (Blundell’s Creek, Brindabella Mts., ACT, Australia) but unfortunately without more precise data. Black cypress pine (*Callitris edlicheri*) could occur at the site, but many species of *Eucalyptus* are also possible candidates as to the host. The host tree remains unknown.

*Schizophilus subrufus* (Randall) breeds in angiosperms [27], as does *Anischia* [24]. *Palaeoxenus dohrnii* (Horn) breeds in gymnosperms (6, Otto, in litt.; Muona, pers. obs.). The host for Phlegoninae species is not known.

Chang et al. [6] and Muona et al. [10] showed that basal lignicolous Eucnemidae diversification had taken place in the Jurassic and the Cretaceous. Here, we demonstrated that Anischiinae were present in the Cretaceous as well. Thus, both Palaeoxeninae breeding in gymnosperms and Anischiinae breeding in angiosperms belong to clades older than the main angiosperm radiation. This dating strongly supports the hypothesis that lignicolous eucnemids first bred in gymnosperms and invaded angiosperms later on. *Anischia* larvae with endodont mandibles would thus represent an adaptive radiation from gymnosperms to angiosperms without major cephalic modifications, whereas *Palaeoxenus* larvae, although breeding in gymnosperms, shares the highly typical head modifications found in more than a thousand extant species developing in angiosperms [26]. This scenario indicates a problem of more general nature. Care should be taken when extrapolating from the structural specializations of modern insect larvae their way of life in distant past. On the other hand, insect larvae developing in wood infested by fungi may succeed with simple-looking morphological structures just well as with complex ones.

The feeding methods used by these beetles may have simplified these evolutionary switches. Eucnemidae use extra-oral digestion, feeding on the bracket-fungus hyphae in the infected wood. In the few cases studied, the fungus species appears to be the deciding factor, not the tree species [5,26] and Muona, unpublished). As many bracket-fungi utilize both angiosperms and gymnosperms as hosts, the potential for a beetle to switch over from one tree species to another one is clear.

Anischiinae and Paleoxeninae represent two early lineages of Eucnemidae. They are separate clades, however, as both groups have solid synapomorphies defining them. Both groups diversified before the main angiosperm radiation in Late Jurassic–Early Cretaceous, but their subsequent histories are very different. Palaeoxeninae, still developing in conifers, survive only in the Southern Californian Mountains [6], whereas Anischiinae, utilizing fungi growing on angiosperms, are found in the tropics of South America, Africa, Asia, and the Pacific [24,28,29]. Eucnemidae clades have been able to find worldwide success utilizing fungi attacking angiosperms, whereas sticking to the coniferous route has turned out to be an evolutionary dead-end.

## Figures and Tables

**Figure 1 insects-12-00105-f001:**
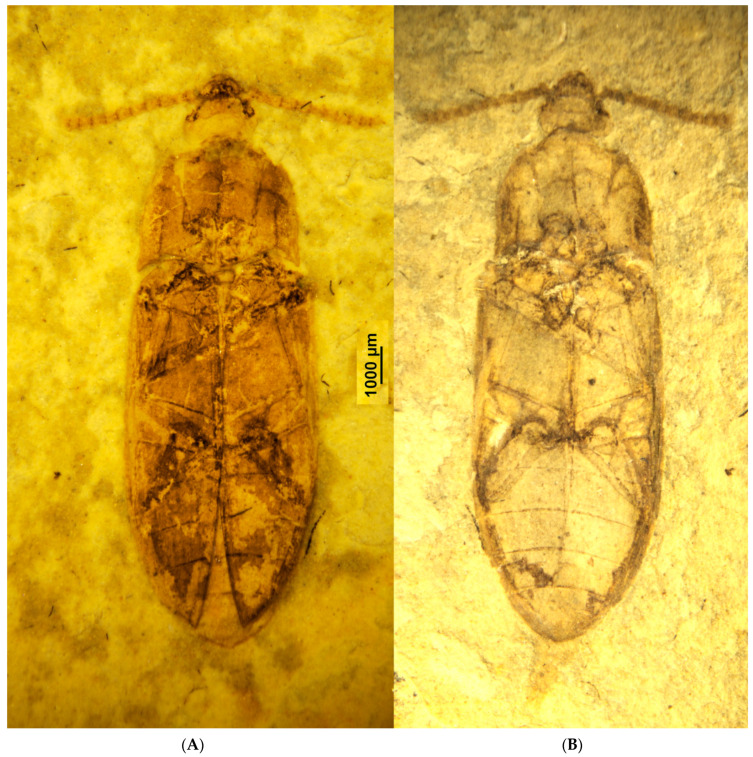
*Rheanischia brevicornis* n.sp. Holotype. (**A**) Dorsal view (2018344), (**B**) ventral view (2018344).

**Figure 2 insects-12-00105-f002:**
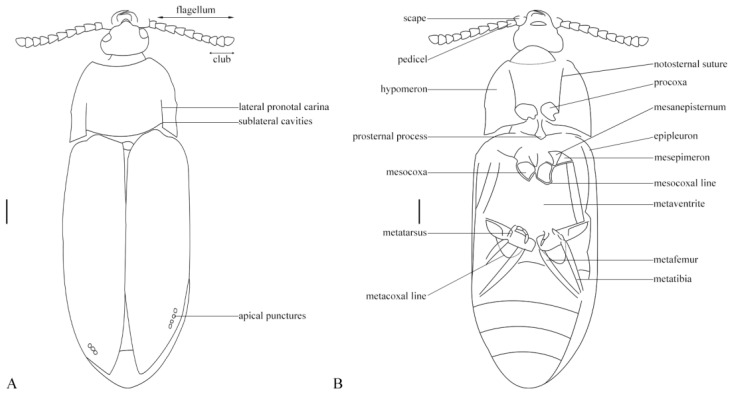
*Rheanischia brevicornis* n.sp. Holotype. Drawing. (2018344). (**A**) Dorsal view, (**B**), ventral view.

**Figure 3 insects-12-00105-f003:**
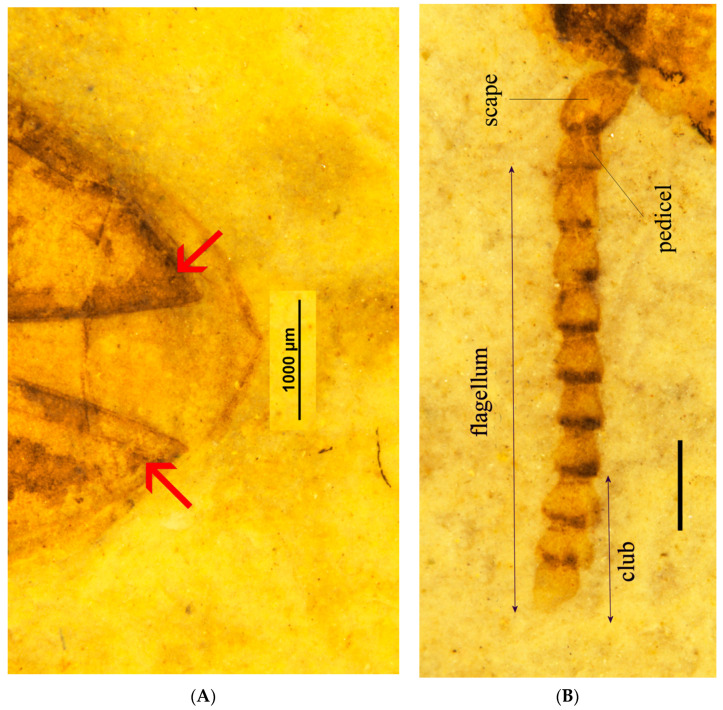
(**A**) *Rheanischia brevicornis* new species., holotype, elytral apices, arrows indicate excretory punctures; (**B**) paratype, antenna.

**Figure 4 insects-12-00105-f004:**
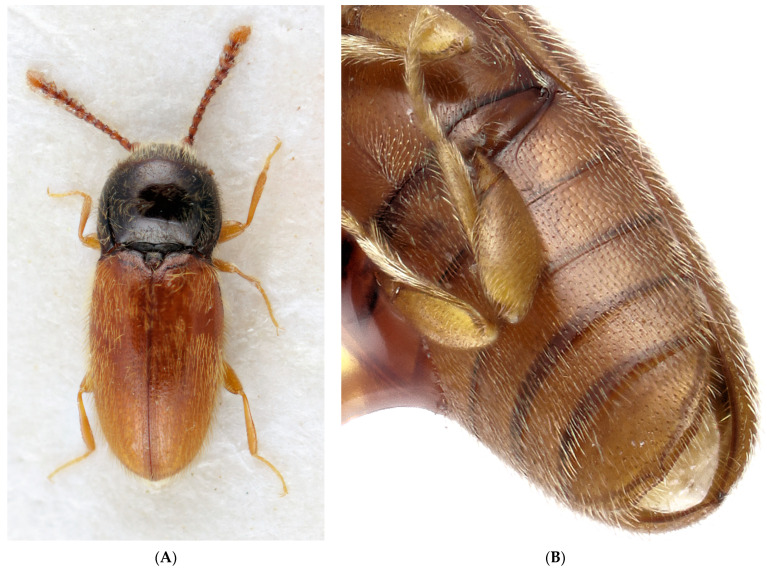
(**A**) *Anischia bicolor* Lawrence, dorsal view, length 2.20 mm; (**B**) *Anischia kuscheli* Lawrence, paratype, ventral view of abdomen showing metacoxa and metacoxal line on right side on first visible ventrite. Note the missing metacoxal plates and free apical abdominal segments; length from left upper corner to end of elytra is 1.03 mm.

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
