# Peer review of "Subfamily Anischiinae (Coleoptera: Eucnemidae) in Early Cretaceous of Northeast China"

_insects, 2021, doi:10.3390/insects12020105_

Round 1
Reviewer 1 Report
see attached file

Author Response
All incorporated
Reviewer 2 Report
Please refer to specific editorial suggestions in the manuscript (attached).

Author Response
All changes incorporated
Reviewer 3 Report
In the manuscript the description of a extint new species and genus are given. The new finding surely are interesting. My main concern is about the dating of the fossil, since the age of the strata is not given for sure (rows 76-79), thus one can suppose that the age of this group cannot be assessed with certainty at present. Perhaps authors should thoroughly discuss the issue.
Furthermore, the names of the anatomical parts are too small in fig 2, thus they must be enlarged.
The description gave enough details, and the proposed taxonomic position of the species seems correct.
Author Response
- We follow the present dating for these deposits
- 2. We agree and have asked for this to take place